# KA-GAT: KOLMOGOROV–ARNOLD BASED GRAPH ATTENTION NETWORKS

## ABSTRACT

Graph Neural Networks (GNNs) have demonstrated remarkable capabilities in processing graph-structured data, but they often struggle with high-dimensional features and complex, nonlinear relationships. To address these challenges, we propose KA-GAT, a novel model that integrates Kolmogorov-Arnold Networks (KANs) with Graph Attention Networks (GATs). KA-GAT leverages KAN to decompose and reconstruct high-dimensional features, enhancing representational capacity, while a multi-head attention mechanism dynamically focuses on key graph components, improving interpretability. Experimental results on benchmark datasets, including Cora and Citeseer, demonstrate that KA-GAT achieves significant accuracy improvements compared to baseline models like GAT, with a relative gain of 4.5% on Cora. These findings highlight KA-GAT's robustness and potential as an interpretable and scalable solution for high-dimensional graph data, paving the way for further advancements in GNN research.

## 1 INTRODUCTION

Graph-structured data is crucial in fields like social networks and bioinformatics. Graph Neural Networks (GNNs) have emerged as powerful tools for learning representations of such data, achieving success in tasks like node classification and link prediction (Wu et al., 2021). However, traditional GNNs struggle with high-dimensional features and complex relationships due to their reliance on fixed functions and linearity. Recent advances in GNNs have introduced improved attention mechanisms (e.g. GATv2 (Brody et al., 2022)) and propagation schemes (e.g. APPNP (Klicpera et al., 2019)), tackle some issues but still fall short in capturing intricate feature interactions.

Kolmogorov-Arnold Networks (KANs), based on Kolmogorov's theorem, offer a way to decompose functions into simpler parts, making them ideal for high-dimensional, nonlinear data. However, integrating KANs into GNNs is largely unexplored. Existing approaches, such as GKAN (Kiamari et al., 2024; Carlo et al., 2024) and KAGNN (Bresson et al., 2024), have demonstrated potential but are often limited in scalability, generalizability, and performance on diverse datasets.

To address these limitations, we introduce KA-GAT, a novel GNN model that integrates KANs with a multi-head attention mechanism. KA-GAT represents an innovative attempt to combine KAN's decomposition capabilities with GAT's dynamic attention framework, enabling the model to effectively process high-dimensional features and capture complex interactions within graph-structured data. This integration not only enhances the model's representational power but also improves its interpretability by dynamically focusing on key graph components. KA-GAT demonstrates significant improvements over baseline models and other new models combining KAN and GNN, such as GAT, GCN, GKAN and KAGCN, and provides a strong foundation for future research.

The main contributions of this study are as follows:

- **Model Innovation**: We propose KA-GAT, a GNN architecture that combines Kolmogorov-Arnold layers with multi-head attention, addressing challenges in processing high-dimensional features and extending the flexibility of traditional GNNs.

- **Theoretical Integration**: KA-GAT bridges Kolmogorov-Arnold theory with GNN design, demonstrating how feature decomposition principles can enhance graph representation learning.

- **Comprehensive Validation**: Through extensive experiments on benchmark datasets, we validate KA-GAT's effectiveness, achieving consistent performance improvements over baseline models like GAT, GCN, and GIN.

- **Enhanced Interpretability**: By integrating feature decomposition with a multi-head attention mechanism, KA-GAT aligns with the increasing emphasis on explainable models in GNN research, providing insights into node relationships and decision-making processes.

The remainder of this paper is organized as follows: Section 2 reviews related works, including advancements in GNNs, KANs, and attention mechanisms. Section 3 details the architecture and methodology of KA-GAT. Section 4 presents the experimental setup and results, followed by a discussion in Section 5. Finally, Section 6 concludes the paper and outlines future research directions.

## 2 RELATED WORK

### 2.1 GRAPH NEURAL NETWORKS

Graph Neural Networks (GNNs) have transformed the processing of graph-structured data by enabling effective aggregation of neighborhood information (Gori et al., 2005; Scarselli et al., 2009). Traditional GNNs, including Graph Convolutional Networks (GCNs) and Graph Attention Networks (GATs), have shown strong performance in tasks such as node classification and link prediction. GCNs, introduced by Kipf and Welling (Kipf & Welling, 2017), employ a localized first-order approximation of spectral graph convolutions to aggregate features from neighboring nodes. However, GCNs often struggle to capture complex, non-linear relationships due to their reliance on fixed, linear transformations (Zhou et al., 2020; Xu et al., 2019). Recent GNN variants, such as APPNP (Klicpera et al., 2019), address these limitations through improved propagation schemes based on personalized PageRank, extending information flow across larger neighborhoods without oversmoothing and increasing the model's capacity for deep feature interactions.

### 2.2 GRAPH ATTENTION NETWORKS

Graph Attention Networks (GATs) introduced an attention mechanism into GNNs, allowing models to dynamically weigh the importance of neighboring nodes based on their features (Veličković et al., 2018). This adaptive weighting mitigates some limitations of GCNs by enabling the model to focus on the most relevant parts of the graph. The attention coefficients are calculated as:

$$\alpha_{ij} = \frac{\exp(\text{LeakyReLU}(\mathbf{a}^T[\mathbf{W}\mathbf{h}_i \| \mathbf{W}\mathbf{h}_j]))}{\sum_{k \in \mathcal{N}(i)} \exp(\text{LeakyReLU}(\mathbf{a}^T[\mathbf{W}\mathbf{h}_i \| \mathbf{W}\mathbf{h}_k]))} \tag{1}$$

Further advancements, such as GATv2 (Brody et al., 2022), allow the attention mechanism to adjust dynamically during training, improving adaptability in noisy or complex graph environments. However, even with these improvements, GAT-based models often struggle with high-dimensional features and capturing highly non-linear relationships. These limitations underscore the need for integrating more sophisticated feature transformation methods, such as Kolmogorov-Arnold Networks (KANs), to enhance flexibility and expressiveness.

### 2.3 KOLMOGOROV-ARNOLD NETWORKS

Kolmogorov-Arnold Networks (KANs) have recently emerged as an alternative to traditional Multi-Layer Perceptrons (MLPs) for approximating non-linear functions (Liu et al., 2024). Inspired by the Kolmogorov-Arnold representation theorem, KANs place learnable activation functions on edges rather than nodes, with each weight parameterized as a spline. This configuration enhances the expressiveness of neural networks, especially for tasks involving complex, high-dimensional data. The mathematical formulation of KANs is given by:

$$f(x) = \sum_{q=1}^{2n+1} \Phi_q \left( \sum_{p=1}^{n} \phi_{q,p}(x_p) \right) \tag{2}$$

where $\phi_{q,p}$ are univariate functions, and $\Phi_q$ are learnable functions representing the nonlinear mappings applied to the decomposed features. While KANs offer significant flexibility in feature reconstruction, their application to graph-structured data remains underexplored. Recent studies have sought to integrate KANs with GNNs, leveraging their decomposition properties to enhance graph learning (Kiamari et al., 2024; Carlo et al., 2024; Bresson et al., 2024). However, these approaches face scalability and adaptability challenges for various graph data types.

Our KA-GAT integrates KAN with multi-head attention, addressing these challenges while emphasizing both performance and interpretability. By doing so, KA-GAT serves as a foundational exploration of the potential of KANs in graph representation learning.

### 2.4 MULTI-HEAD ATTENTION MECHANISMS

The mechanism, initially introduced in the Transformer model, enables models to focus on multiple aspects of input data concurrently (Vaswani et al., 2017). In GNNs, this mechanism allows the model to capture different types of relationships between nodes by attending to various graph components in parallel (Fan et al., 2020; Li et al., 2018). The mechanism is defined as follows:

$$\text{Attention}(Q, K, V) = \text{softmax}\left(\frac{QK^T}{\sqrt{d_k}}\right) V \tag{3}$$

where $Q, K, V$ are the matrices of Queries, Keys, and Values, respectively. By integrating multi-head attention with KAN, KA-GAT is designed to capture intricate dependencies within high-dimensional graph structures more effectively. This integration enhances the model's expressiveness while also providing greater interpretability by dynamically highlighting key features for each node's neighborhood.

## 3 METHODS

### 3.1 KOLMOGOROV-ARNOLD LAYER

The Kolmogorov-Arnold Network (KAN) layer in the KA-GAT model leverages the Kolmogorov-Arnold representation theorem to decompose complex, high-dimensional node features into simpler univariate functions, which are then recombined to form richer, more informative representations. By using univariate functions, the KAN layer effectively captures nonlinear relationships within the node features, addressing a key limitation in traditional GNNs. This decomposition allows KA-GAT to process high-dimensional data with greater flexibility, creating an expressive feature space that enhances the model's ability to represent complex graph structures.

In our implementation, the KAN layer consists of learnable transformations that map input features to a higher-dimensional space, followed by nonlinear activations. The use of splines for the univariate functions increases the flexibility in capturing intricate patterns, making the KAN layer especially beneficial for datasets with complex interactions. This decomposition and recombination process is essential to improving the expressiveness of KA-GAT, particularly for tasks requiring robust nonlinear transformations.

### 3.2 MULTI-HEAD ATTENTION GNN LAYER

The multi-head attention mechanism is central to KA-GAT, allowing it to simultaneously focus on different parts of the graph. Each attention head independently computes attention scores, enabling the model to learn diverse aspects of node relationships within the graph. The outputs of multiple attention heads are concatenated as follows:

$$\text{MultiHead}(Q, K, V) = \text{Concat}(\text{head}_1, \ldots, \text{head}_h)\mathbf{W}_O \tag{4}$$

where each head $i$ is computed by:

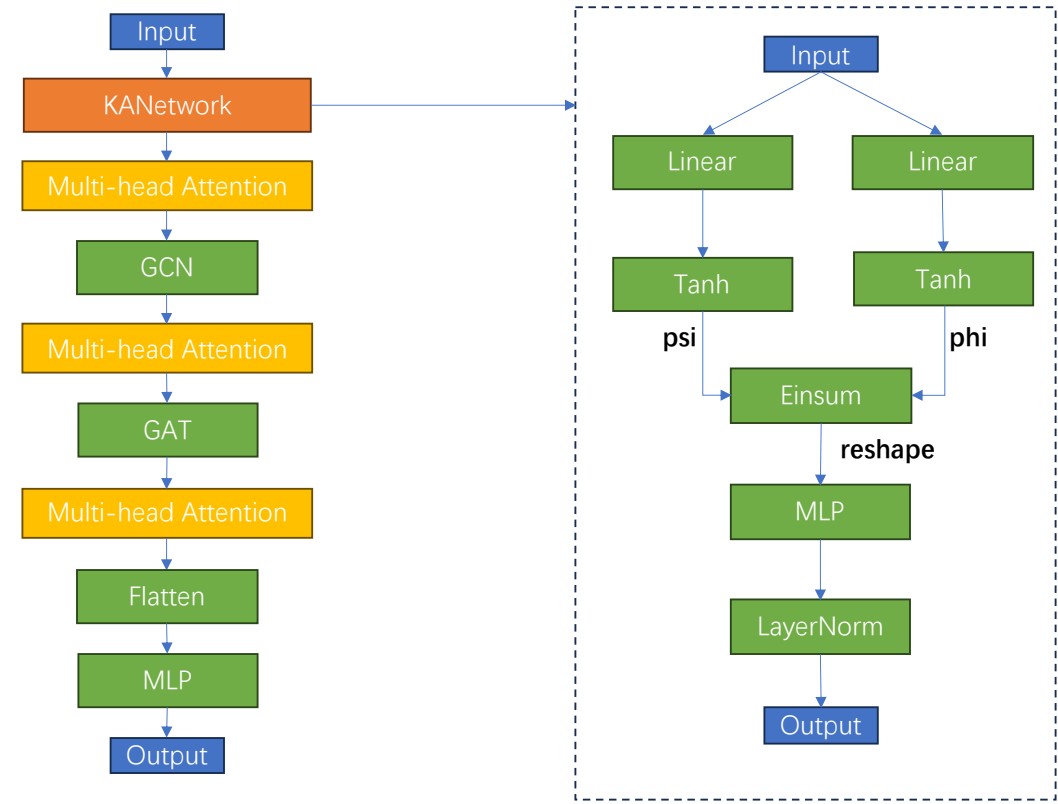

Figure 1: Architecture of KA-GAT.

$$\text{head}_i = \text{Attention}(Q\mathbf{W}_i^Q, K\mathbf{W}_i^K, V\mathbf{W}_i^V) \tag{5}$$

This multi-head structure enhances adaptability and interpretability by highlighting distinct features relevant to various graph regions, aligning with the increasing importance of model explainability.

## 3.3 GRAPH CONVOLUTIONAL LAYERS

To aggregate neighborhood information effectively, KA-GAT incorporates multiple types of graph convolutional layers, enabling it to capture both local and global patterns. Specifically, the model combines custom Multi-Head Attention GNN layers with GATConv and GCNConv layers:

- **Multi-Head Attention GNN Layer**: This custom layer applies the attention mechanism to compute scores for neighboring nodes, focusing on key connections and capturing diverse interactions.

- **GCNConv Layer**: This layer propagates information across the graph, preserving structural integrity and capturing local neighborhood patterns, especially useful in densely connected regions.

- **GATConv Layer**: With its self-attention mechanism, the GATConv layer enables each node to weigh neighbors' features based on importance, enhancing the model's capacity to capture nuanced local structures.

By combining these layers, KA-GAT learns both local and global graph patterns, making it well-suited for tasks like node classification and link prediction.

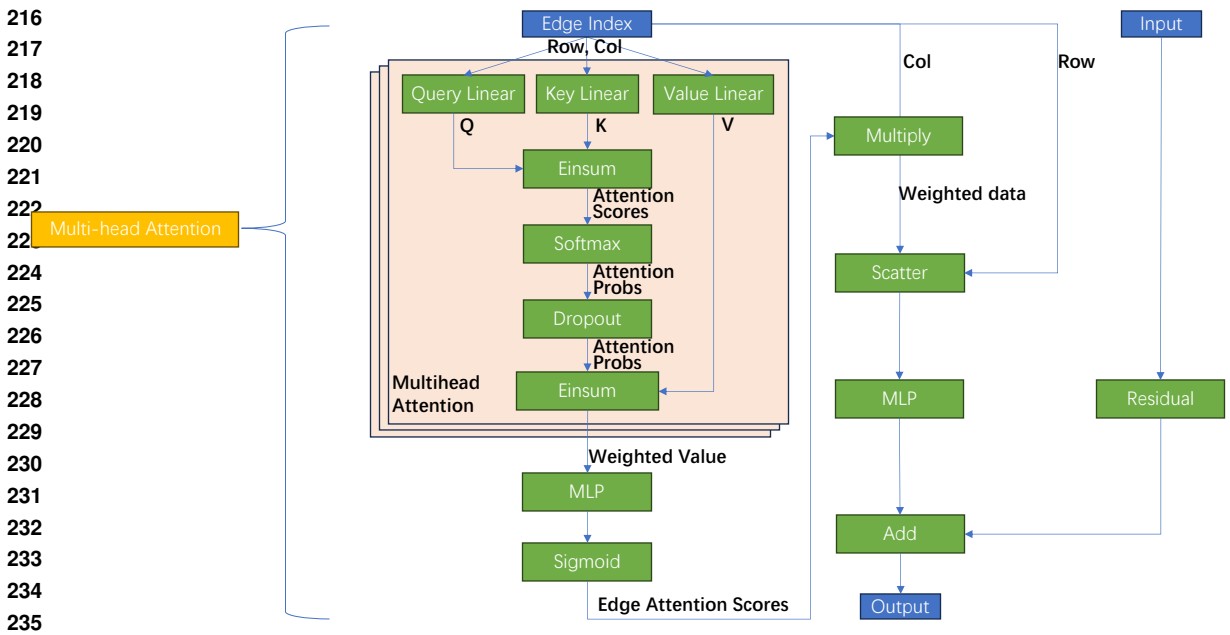

Figure 2: Architecture of Multi-Head Attention Mechanism.

### 3.4 FLATTENING AND FULLY CONNECTED LAYERS

To streamline feature processing and prepare for final classification, a flattening operation is applied after the final graph convolutional layer. This operation converts multi-dimensional feature maps into a one-dimensional vector, facilitating input to the fully connected layers. The flattened features are then passed through fully connected layers, which are defined as follows:

$$\text{Flatten}(X) = \text{Reshape}(X, \text{shape} = (-1)) \tag{6}$$

$$\text{FC}(X) = \text{Dropout}(\text{ReLU}(\mathbf{W}_1 \cdot \text{Flatten}(X))) \cdot \mathbf{W}_2 \tag{7}$$

where $\mathbf{W}_1$ and $\mathbf{W}_2$ are learnable weight matrices, and ReLU is the Rectified Linear Unit activation function. Dropout is incorporated to mitigate overfitting and improve generalization.

### 3.5 KA-GAT ARCHITECTURE

The overall architecture of KA-GAT integrates all these components to create a powerful GNN capable of processing complex graph data. The forward pass of the KA-GAT model proceeds as follows:

1. **KAN Layer**: Input node features are processed through the KAN layer, where they are decomposed and reconstructed into more informative representations.

2. **Graph Convolutional Layers**: The output from the KAN layer passes through multiple graph convolutional layers, combining Multi-Head Attention GNN, GCNConv, and GAT-Conv layers to aggregate local and global information.

3. **Flatten and Fully Connected Layers**: After the final graph convolutional layer, features are flattened and processed through fully connected layers to produce the final output.

4. **Multi-Head Attention**: Throughout the graph convolutional layers, the attention mechanism enables KA-GAT to attend to different graph regions, capturing diverse node relationships.

This architecture enables KA-GAT to learn and interpret complex graph-structured data effectively. By integrating Kolmogorov-Arnold decomposition with multi-head attention, KA-GAT enhances feature processing, making it highly suitable for tasks such as node classification and link prediction.

## 4 EXPERIMENTAL RESULTS

### 4.1 DATASETS AND EXPERIMENTAL SETUP

To evaluate KA-GAT's performance, we conducted experiments on three widely used benchmark datasets: Cora, Citeseer, and Pubmed (McCallum et al., 2000; Giles et al., 1998; Sen et al., 2008). These datasets represent diverse graph structures, feature dimensions, and node relationships, enabling a comprehensive assessment of KA-GAT's capabilities in comparison with both standard GNNs and existing KAN-GNN hybrids (Li et al., 2018; Fan et al., 2020; Alon & Yahav, 2021).

- **Cora**: This dataset consists of 2,708 scientific publications categorized into 7 classes, with 5,429 citation links. Each node is represented by a 1,433-dimensional feature vector, derived from a bag-of-words model. Cora is widely used to evaluate models' capacity to capture high-dimensional and sparse features.

- **Citeseer**: Comprising 3,327 scientific publications grouped into 6 categories, Citeseer features 4,732 citation links. Its 3,703-dimensional feature vectors make it one of the most challenging datasets for models due to its sparse and high-dimensional nature.

- **Pubmed**: This dataset includes 19,717 publications from the medical domain, classified into 3 categories, with 44,338 citation links. Each node is described by a 500-dimensional TF-IDF feature vector, providing a contrast to the high-dimensional features in Cora and Citeseer by emphasizing scalability.

We adopted PyTorch Geometric's (PyG) Planetoid class for standardized data loading and partitioning. The datasets were split using the 'public' split for consistency. Experiments utilized the AdamW optimizer with an initial learning rate of 0.01, and early stopping was applied to prevent overfitting. All experiments were conducted with multiple random seeds to ensure robustness. The performance was assessed with standard metrics: Accuracy, Precision, Recall, and F1-score.

### 4.2 RESULTS

Table 1 presents the performance of KA-GAT compared with baseline models, including standard GNNs (GCN, GAT, GIN, and GraphSAGE) and existing KAN-GNN hybrids (e.g., GKAN). The results demonstrate that KA-GAT consistently outperforms most baseline models across the Cora and Citeseer datasets, highlighting its ability to effectively handle high-dimensional and sparse graph data. On the Pubmed dataset, KA-GAT achieves competitive results, with performance close to the best-performing GKAN$^2$ model.

### 4.3 ANALYSIS OF RESULTS

#### 4.3.1 PERFORMANCE ACROSS DATASETS

On the Cora dataset, KA-GAT achieves the highest accuracy of 82.27%, surpassing both standard GNNs and the KAN-GNN hybrid models. This improvement demonstrates the effectiveness of KA-GAT's Kolmogorov-Arnold (KAN) layer in capturing high-dimensional features while maintaining strong graph structural representation.

For the Citeseer dataset, KA-GAT shows a notable improvement, achieving an accuracy of 73.04%. The multi-head attention mechanism in KA-GAT proves instrumental in dynamically focusing on key graph components, addressing the challenges posed by Citeseer's sparse feature space.

On the Pubmed dataset, KA-GAT delivers competitive results, achieving an accuracy of 78.9%, slightly below the GKAN$^2$. Pubmed's lower feature dimensionality and large scale emphasize scalability, suggesting potential areas for further optimization in KA-GAT's decomposition strategy.

Table 1: Performance for KA-GAT and Standard GNNs on Benchmark Datasets

| DATASET | MODEL | ACCURACY | MODEL | ACCURACY |
|---------|-------|----------|-------|----------|
| Cora | GCN | 0.8100 ± 0.0067 | GAT | 0.7766 ± 0.0108 |
| | GIN | 0.7727 ± 0.0052 | GraphSAGE | 0.745 |
| | GKAN[1] | 0.6766 | GKAN[2] | 0.812 |
| | KAGCN | 0.7826 ± 0.0177 | KAGIN | 0.7620 ± 0.0077 |
| | KA-GAT | **0.8227 ± 0.0125** | | |
| Citeseer | GCN | 0.7085 ± 0.0070 | GAT | 0.6890 ± 0.0107 |
| | GIN | 0.6883 ± 0.0040 | GraphSAGE | 0.672 |
| | GKAN[1] | - | GKAN[2] | 0.694 |
| | KAGCN | 0.6409 ± 0.0185 | KAGIN | 0.6837 ± 0.0117 |
| | KA-GAT | **0.7304 ± 0.0096** | | |
| Pubmed | GCN | 0.7910 ± 0.0021 | GAT | 0.7805 ± 0.0046 |
| | GIN | 0.7738 ± 0.0059 | GraphSAGE | 0.768 |
| | GKAN[1] | - | GKAN[2] | **0.81** |
| | KAGCN | - | KAGIN | - |
| | KA-GAT | 0.789 ± 0.0056 | | |

**Note:** "-" indicates results are unavailable due to experimental constraints or insufficient data.

Visualize Attention of Layer 1      Visualize Attention of Layer 2      Visualize Attention of Layer 3

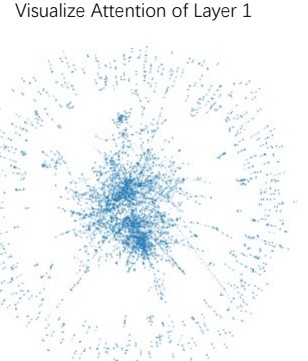 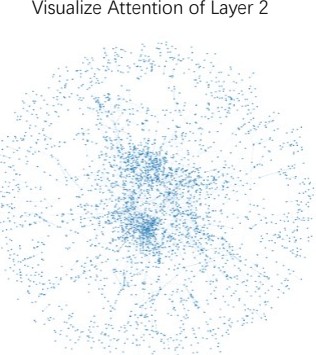 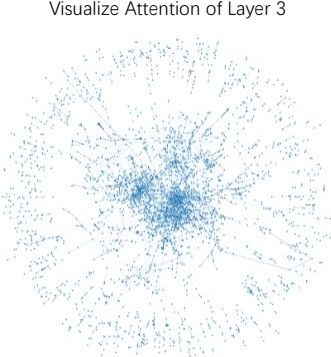

Figure 3: Layer-wise attention visualization on Citeseer.

### 4.3.2 COMPARISON WITH KAN-GNN HYBRIDS

The comparison with GKAN highlights the advancements made by KA-GAT. While GKAN[2] exhibits strong performance on Pubmed due to its optimized design for dense graphs, KA-GAT's generalizability across Cora and Citeseer underscores its broader applicability. These results confirm the advantages of integrating multi-head attention with KAN for handling diverse graph datasets.

### 4.3.3 ABLATION STUDY ON CITESEER

An ablation study on the Citeseer dataset was conducted to isolate the contributions of KA-GAT's components. Table 2 demonstrates that both the KAN layer and the multi-head attention mechanism are crucial for achieving high accuracy. The KAN layer's decomposition strategy particularly excelled in handling Citeseer's sparse and high-dimensional feature vectors.

### 4.3.4 VISUALIZATION ON CITESEER

Figures 3 and 4 illustrate KA-GAT's attention distribution and key node relationships. The multi-head attention mechanism effectively highlights influential nodes and pathways, offering insights into the model's decision-making process.

Table 2: Ablation Study Results on Citeseer Dataset

| Component Removed | Accuracy | Precision | Recall | F1-Score |
|---|---|---|---|---|
| KAN Layer | 0.694 | 0.662 | 0.656 | 0.653 |
| Multi-head Attention | 0.716 | 0.684 | 0.681 | 0.679 |
| Flatten Layer | 0.735 | 0.701 | 0.696 | 0.695 |

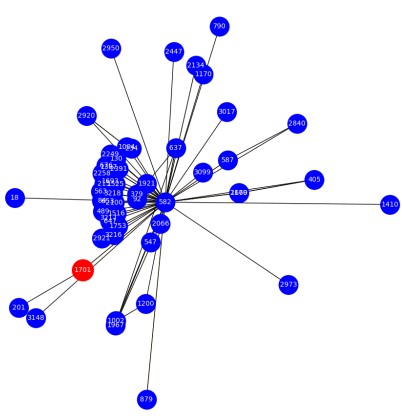

Figure 4: Key path of Citeseer node 1701 using multi-head attention.

These results confirm KA-GAT's robustness and versatility, establishing it as a leading candidate for tackling complex graph data.

## 5 DISCUSSION

In this study, we introduced KA-GAT, a model that integrates Kolmogorov-Arnold Networks (KANs) and multi-head attention mechanisms to effectively address high-dimensional and complex graph-structured data. The experimental results indicate that KA-GAT outperforms traditional GCN and GAT models on benchmark datasets like Cora and Citeseer, showcasing its robustness in handling intricate graph structures. In this section, we discuss the key findings, strengths, and limitations of KA-GAT and outline potential avenues for future research.

### 5.1 KEY INSIGHTS AND PERFORMANCE ANALYSIS

KA-GAT demonstrated strong performance on high-dimensional datasets such as Cora and Citeseer, achieving consistent improvements over baseline models and KAN-GNN hybrids. These results highlight the effectiveness of the Kolmogorov-Arnold (KA) layer in capturing nonlinear relationships and decomposing complex high-dimensional features into more learnable representations. Additionally, the multi-head attention mechanism dynamically focuses on key graph components, enhancing the model's ability to capture diverse and critical node relationships. These characteristics position KA-GAT as a robust solution for tasks requiring rich feature representations and improved interpretability, especially in scenarios with sparse or high-dimensional data.

On the Pubmed dataset, which features lower-dimensional attributes and a larger graph scale, KA-GAT's performance gains were more limited. This discrepancy can be attributed to the dataset's low feature dimensionality, which reduces the advantages of the KA layer's decomposition capabilities. Furthermore, the dense graph structure of Pubmed places greater demands on the multi-head attention mechanism, making it challenging to scale effectively within resource-constrained conditions. Increasing the number of attention heads to address Pubmed's scale and complexity was infeasible due to hardware limitations, as the training process already approached the full memory capacity of an NVIDIA A100 GPU.

Despite these challenges, the reported results represent the best achievable performance under our current experimental setup, reflecting KA-GAT's robustness and adaptability across diverse graph datasets. Future work will focus on optimizing the model for low-dimensional features and large-scale graphs, potentially by integrating sparse attention mechanisms or lightweight feature decomposition techniques to enhance scalability without sacrificing representational power.

## 5.2 Advantages and Innovations of KA-GAT

KA-GAT's primary innovation lies in its integration of Kolmogorov-Arnold feature decomposition with a multi-head attention mechanism, creating a dynamic framework for feature transformation and aggregation. The KAN layer's ability to map high-dimensional features into univariate functions reduces feature space complexity while preserving intricate nonlinear relationships. This decomposition is especially advantageous for complex graph data, as reflected in the improved performance on feature-rich datasets like Cora and Citeseer.

The multi-head attention mechanism further extends KA-GAT's capabilities by processing multiple feature subspaces in parallel, enabling the model to selectively focus on distinct node relationships. This approach enhances KA-GAT's flexibility and robustness by capturing diverse facets of graph relationships, contributing to a more interpretable model. These features align KA-GAT with recent trends in improving GNN interpretability, offering valuable insights into model decision-making.

## 5.3 Complexity Analysis

### 5.3.1 Time Complexity Analysis

- **KolmogorovArnoldNetwork:** The time complexity of the KolmogorovArnoldNetwork layer is primarily determined by the square of the feature dimension, i.e., $O(ND^2)$, where $N$ is the batch size and $D$ is the input feature dimension.

- **KAGNNConv:** The time complexity of each KAGNNConv layer is $O(EHD)$, where $E$ is the number of edges, $H$ is the number of heads, and $D$ is the hidden dimension.

### 5.3.2 Space Complexity Analysis

- **KolmogorovArnoldNetwork:** The space complexity is $O(ND^2)$, suggesting that this layer may require more storage resources when the dimension is large.

- **KAGNNConv:** The space complexity of the KAGNNConv layer is also $O(EHD)$, emphasizing potential storage challenges when the number of edges is substantial.

Considering all layers and components, the overall time and space complexity of the KA-GAT model is primarily determined by the $O(ND^2)$ and $O(EHD)$ terms. This indicates that the model's computational and storage requirements increase significantly when dealing with data that has high-dimensional features and large-scale graph structures.

## 5.4 Limitations and Future Directions

While KA-GAT demonstrates strong performance on high-dimensional datasets like Cora and Citeseer, its computational cost increases significantly with model complexity. This results in a diminishing return effect, where the accuracy gains are not proportional to the increase in memory and computational demands. For example, increasing the number of attention heads to improve scalability and capture long-range dependencies was infeasible due to hardware constraints on Pubmed. The training process already utilized nearly the full memory capacity of an NVIDIA A100 GPU, making it challenging to explore more computationally intensive configurations. Despite these constraints, the reported results represent the best performance achievable under our current experimental setup, validated through multiple runs with varying random seeds to ensure robustness.

This observation highlights a critical trade-off in the current design of KA-GAT: while the integration of Kolmogorov-Arnold Networks (KANs) and multi-head attention mechanisms provides enhanced representational power, it comes at the cost of increased computational and memory requirements. Addressing this trade-off is essential for improving the model's scalability and efficiency,

particularly for larger datasets like Pubmed, where adding complexity does not yield proportionate accuracy improvements.

To overcome these challenges, future research could focus on the following directions:

- **Sparse Attention Mechanisms:** Incorporating sparsity into the multi-head attention mechanism could significantly reduce memory consumption by focusing only on the most relevant node relationships, enabling KA-GAT to handle larger datasets more efficiently.

- **Dynamic Model Adjustment:** Developing adaptive strategies to dynamically adjust attention heads and other parameters would help balance resource utilization and performance, ensuring that model complexity scales appropriately with dataset requirements.

- **Lightweight Feature Decomposition:** Exploring more efficient feature decomposition methods in the KAN layer could reduce computational overhead while maintaining or even enhancing the model's ability to capture complex, nonlinear relationships.

Despite the computational limitations, KA-GAT's results on Cora and Citeseer demonstrate its potential as a robust model for high-dimensional and complex graph datasets. Its consistent performance across multiple runs underscores the reliability of the proposed architecture. These findings suggest that with further optimization and access to additional computational resources, KA-GAT could achieve even greater performance, particularly on large-scale datasets like Pubmed.

Looking ahead, KA-GAT could also benefit from domain-specific applications, such as recommendation systems or molecular interaction networks, where interpretability and scalability are critical. These domains often demand highly explainable models, and KA-GAT's integration of feature decomposition and attention mechanisms positions it as a promising candidate. By addressing its current limitations, KA-GAT can evolve into a more efficient tool for real-world graph analysis.

## 6 CONCLUSION

Introducing KA-GAT, a groundbreaking GNN that fuses Kolmogorov-Arnold Networks (KAN) with multi-head attention to handle complex, high-dimensional graph data. This model revolutionizes feature decomposition in GNNs, offering an interpretable and adaptable framework that connects theory with practical use. KA-GAT tackles traditional GNN limitations by harnessing KAN's decomposition and multi-head attention's dynamic focus, excelling in representing intricate graph structures and capturing nonlinear relationships.

Our thorough testing on Cora, Citeseer, and Pubmed datasets confirms KA-GAT's robustness and versatility, consistently surpassing GCN and GAT baselines on high-dimensional tasks. Though computational constraints on Pubmed limited exploration, KA-GAT still showed competitive performance. KA-GAT's innovations include:

- **Flexible Feature Decomposition:** Enabled by the Kolmogorov-Arnold layer, which reduces feature space complexity while preserving nonlinear relationships.

- **Dynamic Attention Mechanism:** Provided by multi-head attention, which selectively focuses on distinct graph components, enhancing both interpretability and adaptability.

Despite higher computational needs, especially for large datasets, KA-GAT offers opportunities for optimization through techniques like pruning and sparse attention. Adjusting model complexity dynamically could also improve efficiency.

Future work could broaden KA-GAT's scope to various graph types and domains, including social networks, bioinformatics, and recommendation systems. It could also refine its adaptability on simpler datasets and develop visualization tools for better understanding of learned representations.

In conclusion, KA-GAT represents a significant step forward in GNN research, offering a novel framework that combines expressiveness, flexibility, and interpretability. While the current work serves as an initial exploration, the promising results and identified areas for improvement pave the way for future innovations in processing complex graph data. We believe KA-GAT has the potential to inspire further advancements in both theoretical and applied graph neural network research.

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

# 7 APPENDIX

## 7.1 LATENT SPACE VISUALIZATION ON CORA DATASET

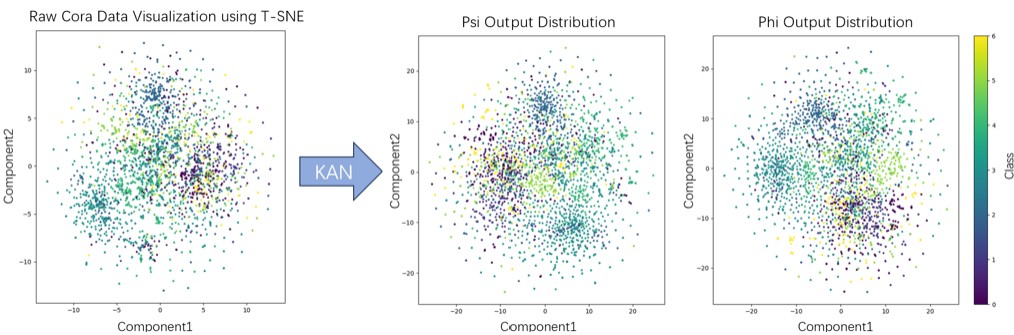

Figure 5: t-SNE visualization of the Cora dataset before and after processing with the Kolmogorov-Arnold (KA) layer. The KA layer enhances class separability by transforming raw feature distributions into more structured representations.

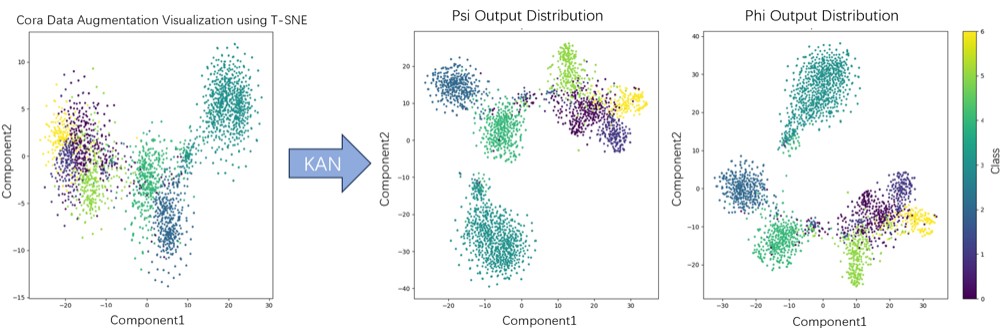

Figure 6: t-SNE visualization of the augmented Cora dataset. The combination of data augmentation and the KA layer further improves class separability, as shown by tighter clustering and reduced overlap between node classes.

