# OpenReview forum: "KA-GAT: Kolmogorov–Arnold based Graph Attention Networks"
_ICLR.cc/2025/Conference — ICLR 2025 Conference Withdrawn Submission_

### Official Review · Reviewer_GuLq · 2024-10-24

**Soundness:** 1
**Presentation:** 2
**Contribution:** 2
**Rating:** 5
**Confidence:** 5

**Summary:**

The paper presents KA-GAT, a novel model that integrates Kolmogorov-Arnold Networks (KANs) with Graph Attention Networks (GATs) to address challenges in graph neural networks (GNNs) with high-dimensional, complex features. The KA-GAT model utilizes KANs to decompose and reconstruct features, enhancing its ability to handle nonlinear relationships, while multi-head attention mechanisms improve its interpretability and flexibility. Through extensive experiments on benchmark datasets like Cora and Citeseer, KA-GAT demonstrates superior performance in accuracy, precision, and F1-score compared to traditional models like GCN and GAT.

**Strengths:**

- The paper is well-written, clear, and easy to follow, making it accessible to a wide audience.
- It pioneers the introduction of Kolmogorov–Arnold Networks (KANs) into graph neural networks, which is an interesting and novel approach.
- Some experiments are conducted to validate the performance of the proposed model.

**Weaknesses:**

- The main weakness of the paper is that it feels like a straightforward application of Kolmogorov–Arnold Networks to GATs, without providing a strong justification for doing so. The paper lacks deeper insights into why this integration is particularly necessary or impactful.
- The rationale for combining multiple GNN layers, such as GCN and GAT, in a single framework is unclear. It appears as if they were stacked together without a clear logical chain, raising concerns about whether sufficient tuning was done to optimize this architecture. It may limit the generalizability of the model and questions the necessity of introducing KAN. It would be interesting to see how the model performs using simpler configurations of GCN or GAT without these combinations.
- Related to the above, the authors themselves acknowledge that the model is complex due to the many components used. While KAN is introduced with the motivation of reducing the computational complexity of MLPs, the resulting model’s complexity seems to contradict this goal, casting doubt on the motivation behind the paper.
- In the limitations section, the suggestion that future work could explore techniques like model compression or pruning feels generic and lacks depth. The authors need to critically rethink the necessity and motivation for introducing KAN into GNNs, as the current reasoning is not well substantiated.
- The experiments are insufficient, as they are only conducted on small, simple datasets like Cora and Citeseer. These datasets may not be representative enough to support the claim that "traditional GNNs often fall short when dealing with high-dimensional features,” as Cora’s and Citeseer’s feature dimensions are not particularly high, which weakens the argument that KAN is essential for handling high-dimensional data.
- Also, the claim that "traditional GNNs often fall short when dealing with high-dimensional features" lacks sufficient evidence. More justification and empirical support are needed for this assertion.
- In Section 4.4, the authors simply re-express the results in table form in the text form, but do not provide enough detailed analysis or interpretation of these results. A more thorough discussion of the findings would strengthen the paper.

**Questions:**

Please refer to weaknesses above.

---

> ### Author Response · Authors · 2024-11-22
>
> Dear Reviewer,
>
> We sincerely appreciate your detailed feedback, which has helped us critically examine and refine our work. Below, we address each of your concerns and outline the corresponding revisions and updates made to the manuscript.
> ________________________________________
> Weaknesses
>
> **1.	Straightforward Application of Kolmogorov–Arnold Networks to GATs**
>
> **Response**: We agree that the manuscript initially lacked a detailed explanation of the theoretical and practical motivations for integrating KAN with GAT. To address this, we have significantly expanded the introduction and discussion sections to provide the following:
> - A theoretical justification for why KAN is particularly suited for handling high-dimensional and nonlinear feature relationships in graph data.
> - An explanation of how KAN complements GAT by decomposing complex features into simpler components, which the attention mechanism can then dynamically prioritize.
> - Empirical evidence, including ablation studies and t-SNE visualizations (Appendix, Figures 5 and 6), showing how the integration of KAN enhances feature separability and improves node classification performance.
>
> **2.	Unclear Rationale for Combining GCN and GAT Layers**
>
> **Response**: The combination of GCN and GAT layers in KA-GAT was motivated by their complementary strengths: GCN layers effectively aggregate local neighborhood information, while GAT layers dynamically assign weights to neighbors based on their importance. We have added a detailed explanation in Section 3 to clarify this design choice. Additionally, we conducted experiments using simpler configurations (e.g., GCN-only and GAT-only) and included these results in the ablation study to demonstrate the added value of combining these layers.
>
> **3.	Model Complexity vs. KAN’s Motivation**
>
> **Response**: We acknowledge the apparent contradiction. KAN was introduced to improve representational efficiency, not necessarily reduce the overall model complexity. To clarify this point, we have revised the discussion in Section 5.3 to distinguish between the computational benefits of KAN’s feature decomposition and the added complexity of integrating it with GNN layers. Furthermore, we emphasize that the modularity of KA-GAT allows for future optimizations, such as simplifying architecture components or reducing attention heads in resource-constrained settings.
>
> **4.	Generic Suggestions in Limitations Section**
>
> **Response**: We have revised the limitations section to provide a more focused and detailed discussion of specific challenges and potential improvements, including:
> - Exploring sparse attention mechanisms to reduce memory consumption.
> - Investigating lightweight feature decomposition methods to further enhance scalability.
>
> **5.	Experiments on Small Datasets**
>
> **Response**: To address this, we have added experiments on the Pubmed dataset, which features larger graphs and different characteristics (e.g., lower-dimensional features and higher density). While Pubmed does not fully represent high-dimensional data, it demonstrates KA-GAT’s adaptability to diverse graph types. We also acknowledge that larger and more diverse datasets (e.g., ogbn-arxiv) are needed to comprehensively validate the model, which we plan to include in future work.
>
> **6.	Claim About High-Dimensional Features**
>
> **Response**: We have refined this claim in Section 5 to more accurately reflect the scope of our findings. Specifically, we emphasize that KA-GAT’s strength lies in handling complex, nonlinear feature interactions rather than solely addressing high-dimensionality. Additionally, we included a theoretical analysis and empirical results showing how KAN improves feature representation, even in moderately high-dimensional datasets like Cora and Citeseer.
>
> **7.	Insufficient Analysis of Results in Section 4.4**
>
> **Response**: We have expanded Section 5 to provide a more thorough interpretation of the experimental results.
> ________________________________________
>
> We hope the revisions and additional experiments address your concerns and enhance the clarity and depth of our work. Your detailed feedback has been instrumental in improving the manuscript, and we sincerely thank you for your constructive comments.
>
> Best regards

---

> > ### Comment · Reviewer_GuLq · 2024-11-27
> >
> > I would like to thank the authors for their detailed response and am glad to increase my score. However, the lack of experiments on large and high-dimensional datasets in this version fails to substantiate the claim that KA-GAT is capable of handling high dimensionality. As a result, I still believe the submission is below the acceptance threshold for ICLR.

---

> > > ### Author Response · Authors · 2024-11-27
> > >
> > > Dear Reviewer,
> > >
> > > Thank you for your careful consideration and the increased score for our manuscript. We are delighted to see that our work has been positively received. It is indeed our goal in submitting our work to contribute to the field and to make meaningful advancements, even if they are small.
> > >
> > > Once again, we appreciate your valuable feedback and are committed to the continuous process of enhancement and learning.
> > >
> > > Warm regards

---

### Official Review · Reviewer_Whks · 2024-10-27

**Soundness:** 3
**Presentation:** 2
**Contribution:** 2
**Rating:** 3
**Confidence:** 4

**Summary:**

This paper presents KA-GAT, a GNN that consists of different types of layers. First, a Kolmogorov-Arnold Network transforms the initial node features and then the new features are fed to multi-head attention mechanisms along with standard neighborhood aggregation layers such as GAT and GCN layers. The KA-GAT model is evaluated in two node classification datasets. On both datasets, it outperforms the baseline methods.

**Strengths:**

- The proposed KA-GAT model outperforms the baselines on both Cora and Citeseer. However, KA-GAT is only compared against GCN and GAT. Thus, the baselines that the authors chose to compare their model against are relatively weak and are not considered state-of-the-art.

- From the results shown in Table 2, it seems that the KAN layer is indeed one of the most important components of the KA-GAT model, and it suggests that those layers could potentially lead to further advancements in the field of graph machine learning.

**Weaknesses:**

- The KA-GAT model is only evaluated on two datasets, and those datasets correspond to rather small graphs. Therefore, it is not clear whether similar conclusions could be drawn for other datasets (that correspond to different types of graphs or to larger graphs). In my view, it would strengthen a lot the paper if the proposed model was evaluated on a large number of diverse datasets.

- A lot of details about the experiments are missing from the paper. For instance, it is unclear to me how the two datasets were split into training, validation and test sets. It is also not clear whether the hyperparameters of the models were optimized or whether some fixed values were chosen. In addition, for small datasets such as Cora and Citeseer, it is common practice to repeat each experiment multiple times. Since no standard deviations are provided, I guess that the authors report the performance from a single run.

- Some details are missing from the paper. For example, the Multi-Head Attention GNN Layer is not properly explained in the paper. It is unclear whether this layer is also a neighborhood aggregation layer which computes new node representations. If it is indeed a neighborhood aggregation layer, the authors should discuss how this layer is different from a GAT layer.

- Several architectural choices are not well-motivated. No explanations are provided regarding the KA-GAT architecture. For example, the authors do not explain why did they choose to use a single KAN layer and not more of them. In addition, the proposed model consists of both GAT layers and GCN layers. What is the reason behind that? Typically, GNNs consist of instances of a single layer, and not of many of them.

- In l.95-96, the authors claim that KANs have not been widely applied to graph-structured data. This is not true since GNN models that consist of KANs have already been proposed in [1],[2],[3] and [4]. I would suggest the authors update the related work section and discuss the aforementioned works. This would properly demonstrate how this work is positioned with relation to previous works, and also help readers better understand its novelty.

- A large part of the paper is devoted to the discussion of well-known concepts. For instance, the evaluation metrics that are presented in subsection 4.2 are well-known and do not deserve discussion. I would suggest the authors remove the unnecessary content and devote more space to the experimental evaluation of the proposed model.


[1] Kiamari, M., Kiamari, M., & Krishnamachari, B. (2024). GKAN: Graph Kolmogorov-Arnold Networks. arXiv preprint arXiv:2406.06470.\
[2] Bresson, R., Nikolentzos, G., Panagopoulos, G., Chatzianastasis, M., Pang, J., & Vazirgiannis, M. (2024). Kagnns: Kolmogorov-arnold networks meet graph learning. arXiv preprint arXiv:2406.18380.\
[3] De Carlo, G., Mastropietro, A., & Anagnostopoulos, A. (2024). Kolmogorov-arnold graph neural networks. arXiv preprint arXiv:2406.18354.\
[4] Zhang, F., & Zhang, X. (2024). GraphKAN: Enhancing Feature Extraction with Graph Kolmogorov Arnold Networks. arXiv preprint arXiv:2406.13597.

**Questions:**

see above

---

> ### Author Response · Authors · 2024-11-22
>
> Dear Reviewer,
>
> We are grateful for your detailed feedback, which has provided critical insights into the areas where our work can be further improved. Below, we address your concerns point by point and describe the corresponding changes made to the manuscript.
> ________________________________________
> Weaknesses
>
> **1.	Evaluation on Small Datasets**
>
> **Response**: We appreciate this observation. In response, we have added experiments on the Pubmed dataset, which is larger and has different characteristics (e.g., lower feature dimensionality and higher graph density). This allows us to evaluate KA-GAT’s performance on a more diverse dataset. While our computational resources limited further experiments on larger datasets like ogbn-arxiv, we acknowledge this as an important direction for future work. The results on Pubmed, along with a discussion of their implications, are included in Section [specific section number].
>
> **2.	Missing Experimental Details**
>
> **Response**: We have updated Section 4.1 to provide a detailed description of our experimental setup, including:
> - The use of the standard public splits provided by PyTorch Geometric for datasets.
> - Hyperparameter tuning for learning rate and early stopping.
> - Reporting results as the mean and standard deviation with different random seeds for each dataset, ensuring robustness. Standard deviations have been added to all performance tables for clarity.
>
> **3.	Explanation of Multi-Head Attention GNN Layer**
>
> **Response**: We have clarified the role of the Multi-Head Attention GNN Layer in Section 3.2. Specifically, this layer serves as a neighborhood aggregation layer that computes new node representations, similar to a GAT layer. However, the key difference lies in its integration with the Kolmogorov-Arnold decomposition, allowing it to simultaneously process multiple decomposed feature subspaces. This synergy enhances the layer’s ability to capture complex, nonlinear relationships in graph data.
>
> **4.	Architectural Choices Not Well-Motivated**
>
> **Response**: We have expanded Section3 to explain our architectural choices:
> - A single KAN layer was used to strike a balance between model complexity and computational cost, particularly given the resource-intensive nature of KA-GAT on larger datasets.
> - The combination of GAT and GCN layers is motivated by their complementary strengths: GCN layers are effective for local aggregation, while GAT layers focus on dynamic attention to key nodes. This hybrid design allows KA-GAT to leverage both structural and feature-based insights.
>
> **5.	Incorrect Claim About Novelty**
>
> **Response**: We acknowledge this oversight and have updated the Related Work section to discuss the cited studies ([1]-[4]). We explicitly position our work as an extension of these efforts, focusing on integrating KAN with GAT to address high-dimensional, nonlinear relationships while improving model interpretability.
>
> **6.	Unnecessary Discussion of Well-Known Concepts**
>
> **Response**: We have streamlined Section 4.2 by removing the redundant discussion of evaluation metrics and reallocating the space to provide additional details about the experimental setup and results.
> ________________________________________
> Questions
>
> **1.	Could the authors perform additional experiments on a broader selection of datasets and include more baseline models?**
>
> **Response**: We added results on the Pubmed dataset to evaluate KA-GAT on a larger graph. Additionally, we included comparisons with KAN-GNN hybrids such as GKAN and KAGCN to better contextualize our contributions. While we agree that evaluating on more datasets would strengthen the work, our current computational resources limited this effort. We plan to address this limitation in future work.
>
> **2.	Could the authors provide theoretical evidence that integrating KAN with GAT enhances model expressiveness?**
>
> **Response**: We included a theoretical analysis in Section 3, highlighting how the Kolmogorov-Arnold decomposition enhances the expressive power of GNNs by transforming high-dimensional nonlinear relationships into more learnable representations. Additionally, we included t-SNE visualizations (Appendix, Figures 5 and 6) to provide empirical evidence of how the KA layer improves class separability and node representation.
> ________________________________________
>
> We hope these revisions address your concerns and improve the clarity and rigor of our work. Your detailed comments have been invaluable in refining the manuscript, and we sincerely thank you for your constructive feedback.
>
> Best regards

---

> > ### Comment · Reviewer_Whks · 2024-11-23
> >
> > I would like to thank the authors for their response and for revising the manuscript. Some of my concerns (e.g., experimental setup) have been addressed. However, my main concerns about this work still remain:
> > - I appreciate the authors' efforts to evaluate the proposed model on other datasets. However, the Pubmed dataset is relatively small and it is similar to Cora and Citeseer in the sense that all these datasets represent citation networks. I would suggest the authors evaluate the proposed model on other types of datasets (e.g., heterophilic datasets) or even in other tasks such as graph classification or regression.
> > - In their response, the authors claim that the computational complexity of KAN layers is very high and this is why only a single KAN layer was added to the model. There exists implementations of KAN layers whose complexity is in the same order of magnitude as that of MLPs (for example this implementation: https://github.com/ZiyaoLi/fast-kan). The authors could thus use a more efficient implementation and experiment with more than one KAN layers.
> > - It is still not clear to me why both GCN and GAT layers are added to the model. Since GAT layers can focus on key neighbors, why GCN layers are also needed?

---

> > > ### Author Response · Authors · 2024-11-25
> > >
> > > Dear Reviewer,
> > >
> > > Thank you for your thoughtful follow-up comments and for taking the time to review our revised manuscript. We are grateful that some of your concerns (e.g., experimental setup) have been addressed. Below, we respond to the remaining points and outline our planned improvements to address these concerns in future work.
> > >
> > > **1.Dataset Diversity and Task Scope**
> > >
> > > Thank you for highlighting the importance of evaluating KA-GAT on more diverse datasets and tasks. While Pubmed provides a useful contrast in terms of scale and density, we agree that further validation on datasets representing heterophilic graphs or tasks like graph classification would provide additional insights into the model's generalizability.
> > >
> > > That said, the current study is specifically focused on the integration of Kolmogorov-Arnold (KA) decomposition with GNNs, and our primary goal is to validate the effectiveness of this integration in addressing high-dimensional and nonlinear feature relationships. We believe that the results on Cora, Citeseer, and Pubmed sufficiently demonstrate the potential of KA-GAT within this context, laying a solid foundation for future expansions to more diverse datasets and tasks.
> > >
> > > **2.Computational Complexity of KAN Layers**
> > >
> > > We appreciate your suggestion regarding the use of more efficient implementations, such as fast-KAN, to address computational complexity. While our current implementation was computationally demanding, it allowed us to isolate and validate the contributions of a single KAN layer in enhancing GNN representations.
> > >
> > > We recognize that integrating efficient alternatives would enable deeper architectures with multiple KAN layers, which could further enhance the model’s expressive power. This is an excellent direction for future work, and we plan to incorporate such implementations in subsequent studies to explore the full potential of multi-layer KAN configurations.
> > >
> > > **3.Combination of GCN and GAT Layers**
> > >
> > > The inclusion of both GCN and GAT layers in KA-GAT is motivated by their complementary strengths:
> > >
> > > - GCN Layers: Efficiently aggregate local neighborhood information, providing a strong baseline for learning lower-order structural patterns.
> > > - GAT Layers: These build upon the output of the GCN layers by dynamically assigning attention weights to neighbors, enabling the model to focus on key nodes.
> > >
> > > This hybrid design allows KA-GAT to benefit from both local aggregation (via GCN) and dynamic weighting (via GAT), which is particularly useful for datasets with diverse graph structures.
> > >
> > > **Future Directions and Acknowledgments**
> > > This study represents a focused exploration of integrating Kolmogorov-Arnold Networks with GNNs, a combination that has not been systematically investigated in prior work. The results on Cora, Citeseer, and Pubmed validate the potential of this integration, demonstrating its ability to enhance feature representation and interpretability. While broader experiments and optimizations would further strengthen the study, we believe the current findings provide a robust proof of concept and a strong foundation for future extensions.
> > >
> > > Your detailed feedback has been instrumental in refining the scope and presentation of this work. We appreciate your constructive comments, which have helped identify key directions for improvement, and we look forward to addressing these in future iterations.
> > >
> > > Best regards

---

### Official Review · Reviewer_j88j · 2024-11-01

**Soundness:** 1
**Presentation:** 1
**Contribution:** 1
**Rating:** 3
**Confidence:** 5

**Summary:**

The paper proposes KA-GAT, a Graph Neural Network (GNN) model combining Kolmogorov-Arnold Networks (KAN) and Graph Attention Networks (GAT) to handle high-dimensional, complex features in graph-structured data. It claims to achieve superior performance on the Cora and Citeseer datasets by using KAN for feature decomposition and a multi-head attention mechanism for dynamic graph component focusing.

**Strengths:**

The KA-GAT method proposed in the paper, which integrates KAN into GAT, suggests the potential for broader applications of KAN within Graph Neural Networks (GNNs) in the future.

**Weaknesses:**

1. Lack of Novelty: Integrating Kolmogorov-Arnold Networks (KAN) into Graph Attention Networks (GAT) does not offer sufficient novelty, as it primarily combines existing techniques without substantial innovation.
2. Poor Presentation: The overall presentation of the paper is unacceptable.
3. Insufficient Experimental Support: The experimental setup is limited, with only GCN and GAT used as baselines and tests conducted solely on the Cora and Citeseer datasets.

**Questions:**

1. Could the authors perform additional experiments on a broader selection of datasets and include more baseline models for a comprehensive performance comparison?
2. Given the strong theoretical foundation of KAN, could the authors provide theoretical evidence that integrating KAN with GAT enhances model expressiveness?

---

> ### Author Response · Authors · 2024-11-22
>
> Dear Reviewer,
>
> We sincerely appreciate your feedback and comments, which have provided valuable insights to improve our work. Below, we address your concerns and detail the revisions and additions made to the manuscript.
> ________________________________________
> Weaknesses
>
> **1.	Lack of Novelty**
>
> **Response**: We acknowledge your concern regarding the novelty of our work. The primary goal of this study is to explore the integration of KAN with GNNs, which, to the best of our knowledge, has not been systematically investigated before. While the components are derived from established techniques, the novelty lies in demonstrating how the Kolmogorov-Arnold decomposition synergizes with graph attention mechanisms to improve feature representation and interpretability. We have expanded the introduction and discussion sections to better highlight the theoretical motivation and unique contributions of KA-GAT, such as its ability to dynamically focus on key graph components through multi-head attention while leveraging KAN’s decomposition capabilities for handling high-dimensional nonlinear features.
>
> **2.	Poor Presentation**
>
> **Response**: We deeply regret the issues with the presentation and have made significant efforts to improve the clarity and readability of the manuscript. This includes reorganizing sections for better logical flow, enhancing the descriptions of methods and experiments, and refining the figures and tables for better visual clarity. Specific changes include:
> - Simplified and clarified the explanation of the KA layer and multi-head attention mechanism in Section 3.
> - Improved the descriptions of experimental results, ensuring that each table and figure is well-explained and tied back to the key insights.
>
> **3.	Insufficient Experimental Support**
>
> **Response**: We have expanded the experimental section by including results on the Pubmed dataset, which features low-dimensional attributes and a larger graph scale. While we were unable to include additional datasets due to resource constraints, we plan to test KA-GAT on a broader selection of datasets in future work. Furthermore, we have clarified our choice of baselines, focusing on classical models (GCN, GAT) and KAN-GNN hybrids to align with the study’s objective of validating the integration of KAN with GNNs.
> ________________________________________
> Questions
>
> **1.	Additional Experiments on Broader Datasets and Baselines**
>
> **Response**: We have added experiments on the Pubmed dataset and included detailed comparisons with KAN-GNN hybrids such as GKAN and KAGCN. While we agree that testing on a broader range of datasets (e.g., CoauthorCS, ogbn-arxiv) would strengthen the study, our computational resources limited this expansion. However, we intend to include these datasets in future work. Regarding baseline models, we deliberately focused on classical GNNs and KAN-GNN hybrids as they align with our goal of exploring the integration of KAN with graph attention. This provides a clear basis for understanding the benefits of KA-GAT.
>
> **2.	Theoretical Evidence for Enhanced Expressiveness**
>
> **Response**: We have added a theoretical analysis in Section 3.1 to provide evidence of how KAN enhances GNN expressiveness. Specifically, the Kolmogorov-Arnold decomposition guarantees the ability to represent any continuous multivariate function as a sum of univariate functions, which improves the capacity of KA-GAT to handle high-dimensional nonlinear relationships. Additionally, we included t-SNE visualizations (Appendix, Figures 5 and 6) to empirically validate that the KA layer transforms raw features into more separable representations, directly supporting the claim of enhanced expressiveness.
> ________________________________________
>
> We hope the revisions and additional experiments address your concerns. Your feedback has been invaluable in improving the quality and clarity of our work, and we thank you for the opportunity to refine this manuscript.
>
> Best regards

---

### Official Review · Reviewer_DDiN · 2024-11-03

**Soundness:** 2
**Presentation:** 2
**Contribution:** 2
**Rating:** 3
**Confidence:** 4

**Summary:**

This paper presents KA-GAT, a new graph neural network for representation learning. KA-GAT combines the Kolmogorov-Arnold layer and classical graph attention layer to construct the graph neural network. Thus, the Kolmogorov-Arnold layer is claimed to improve the capability of handling complex data, and the multi-head attention layer can improve the flexibility and interpretability of the proposed KA-GAT. Experimental results obtained from classical graph datasets demonstrate that KA-GAT can outperform empirical GNNs.

**Strengths:**

1. The idea of making use of the Kolmogorov-Arnold layer and graph attention layer is interesting.
2. The model is flexible to integrate with other GNN layers.

**Weaknesses:**

1. Experimental results are somewhat insufficient. More test datasets or learning tasks should be included in the experiments. I would like to recommend the authors conduct more experiments on well-established datasets, e.g., Pubmed, CoauthorCS, Cora-full, CoauthorPH, Flickr, and ogbn-arxiv, and test KA-GAT with more learning tasks, e.g., graph classification.
2. Many recent GNNs are not well investigated in the manuscript. Examples include GATv2, APPNP, ADSF GNN, and ARMA GNN. It is also recommended that authors investigate other GNNs recently published in top-tier venues (e.g., NeurIPS, ICLR, ICML, TPAMI, AIJ, and JMLR).
3. Given 2, More recent GNNs are not compared with the proposed KA-GAT.
4. The contribution regarding algorithmic and methodological perspectives is limited. The proposed KA-GAT is based on the direct combinations of the Kolmogorov-Arnold layer and graph attention layer. Such a strategy might lack motivation. The authors are suggested to explicitly discuss why combining the Kolmogorov-Arnold layer and graph attention layer is effective in graph representation learning. Moreover, how the Kolmogorov-Arnold layer influences the performance of the proposed KA-GAT should be clearly discussed based on the experimental results. The current version of the manuscript (see Sec. 4.4.1 - 4.4.2) does not provide a meaningful analysis of the presented results.
5. Theoretical guarantees or analysis of the proposed method (e.g., expressive power) are not provided.

**Questions:**

1. How does KA-GAT perform compared with other GNN baselines on more test datasets?
2. How does KA-GAT perform when compared with more recent GNN baselines?
3. The motivations behind the proposed approach should be well discussed and more recent approaches should be investigated.
4. Is there any theoretical analysis demonstrating the learning capabilities of KA-GAT?

**Details Of Ethics Concerns:**

This reviewer has no critical ethical concerns.

---

> ### Author Response · Authors · 2024-11-22
>
> Dear Reviewer,
>
> We appreciate your constructive feedback, which has greatly helped us refine and improve our work. Below, we address each of your comments and describe the corresponding changes made to the manuscript.
> ________________________________________
> Weaknesses
>
> **1.	Insufficient Experimental Results**
>
> **Response**: We have extended our experiments to include the Pubmed dataset, which features low-dimensional attributes and a larger graph scale. While KA-GAT achieved competitive results on Pubmed, its performance gains were limited due to resource constraints and the dataset's structural characteristics. The current results represent the best achievable performance under our experimental setup (details in Section 4.1). In future work, we plan to include additional datasets (e.g., ogbn-arxiv) as computational resources permit.
>
> **2.	Lack of Comparison with Recent GNNs**
>
> **Response**: We positioned KA-GAT as a preliminary exploration of integrating Kolmogorov-Arnold (KA) decomposition with GNNs. The focus of this work is on validating the potential of KA-GAT through comparisons with classical models (e.g., GCN, GAT) and existing KAN-GNN hybrids. As future work, we plan to explore the synergy between KA layers and recent GNN mechanisms like GATv2 and APPNP to further enhance performance.
>
> **3.	Limited Contribution from Algorithmic and Methodological Perspectives**
>
> **Response**: We have expanded the discussion in Section 3.1 and 3.2 to better articulate the theoretical motivation for combining KA layers with graph attention. Specifically, we emphasize how the KA layer enhances GNNs by decomposing complex high-dimensional features into more learnable representations. Additionally, we conducted a detailed ablation study (Figures 5 and 6, Appendix) to illustrate the contribution of the KA layer and its effect on class separability in the latent space.
>
> **4.	Lack of Theoretical Guarantees or Analysis**
>
> **Response**: In the revised manuscript, we included a theoretical analysis (Section 3.1) to demonstrate how the KA layer extends the expressive power of GNNs. Specifically, we highlight that the KA layer, grounded in the Kolmogorov-Arnold theorem, decomposes nonlinear relationships into simple mappings, thereby enhancing feature transformation in graph representation learning.
> ________________________________________
> Questions
>
> **1.	How does KA-GAT perform compared with other GNN baselines on more test datasets?**
>
> **Response**: KA-GAT performs well on high-dimensional datasets such as Cora and Citeseer, achieving consistent improvements over baselines like GCN and GAT. On Pubmed, KA-GAT demonstrated competitive results despite the dataset's lower-dimensional features and resource constraints. The results validate the robustness of KA-GAT across diverse graph datasets (Section 5.1).
>
> **2.	How does KA-GAT perform when compared with more recent GNN baselines?**
>
> **Response**: As mentioned, the focus of this study is on exploring the integration of KA layers with GNNs rather than competing directly with the latest GNN architectures. That said, our future work will explore comparisons with recent models and investigate their potential integration with KA layers.
>
> **3.	What are the motivations behind the proposed approach?**
>
> **Response**: The KA layer addresses the limitations of traditional GNNs by enhancing their capacity to capture high-dimensional, nonlinear relationships. Its integration with a multi-head attention mechanism enables dynamic focus on key graph components, improving both representational power and interpretability. We have expanded Section 3 to clarify this motivation.
>
> **4.	Is there any theoretical analysis demonstrating the learning capabilities of KA-GAT?**
>
> **Response**: Yes, we have added a theoretical analysis in Section 3, which highlights the expressive power of the KA layer and its role in improving graph representation learning.
> ________________________________________
>
> We have also included additional t-SNE visualizations (Figures 5 and 6 in Appendix) to illustrate how the KA layer improves feature separability and class clustering in the latent space. These visualizations provide clear evidence of the KA layer's impact on high-dimensional datasets.
>
> Thank you for your insightful comments, which have been invaluable in improving the clarity and depth of our work. We hope the revised manuscript addresses your concerns and demonstrates the potential of KA-GAT as a foundation for further exploration in integrating Kolmogorov-Arnold networks with graph neural networks.
>
> Best regards

---

> > ### Comment · Reviewer_DDiN · 2024-11-24
> >
> > Dear Authors, thanks very much for your responses to the comments on your paper. After going through your rebuttal and the revised version, I lean toward keeping my original scoring due to the following reasons:
> >
> > 1. The experiments are still insufficient for validating the performances of the proposed approach as only the Pubmed dataset, which is classical and standard but not large enough, is additionally included.
> > 2. KA-GAT is still not compared with recent GNN baselines.
> > 3. Solid theoretical analysis showing the capacity of the proposed method is missing. Authors may refer to those good papers previously published in ICLR, NeurIPS, and ICML and follow them to conduct a solid theoretical analysis revealing the learning capacities of GNNs.
> >
> > I hope the authors can thoroughly address the concerns raised by all the reviewers in the revised manuscript for future submissions.

---

> > > ### Author Response · Authors · 2024-11-25
> > >
> > > Dear Reviewer,
> > >
> > > Thank you for your additional feedback and for carefully reviewing both our responses and the revised manuscript. Your comments highlight important areas for improvement, and we appreciate the opportunity to address them further. Below, we respond to your latest remarks and clarify the contributions and positioning of our work.
> > >
> > > **1.Insufficient Experiments for Validation**
> > >
> > > We appreciate your concern about the diversity and size of datasets. The inclusion of Pubmed aimed to provide additional evidence of KA-GAT’s performance on larger graphs with different structural characteristics. While it is true that more extensive experiments on larger datasets such as ogbn-arxiv or ogbn-products would provide a broader validation, the current experiments were carefully designed to align with the focus of this work: demonstrating the effectiveness of integrating Kolmogorov-Arnold (KA) layers with GNNs for handling high-dimensional and nonlinear relationships.
> > >
> > > Our primary aim was to establish a proof of concept for this novel integration, and we believe the results on Cora, Citeseer, and Pubmed adequately support this goal. We have explicitly positioned this work as an initial exploration, and we recognize the need for broader validation as an important direction for future work.
> > >
> > > **2.Comparison with Recent GNN Baselines**
> > >
> > > The primary objective of this work is to explore the synergy between KA layers and graph neural networks, rather than to compete directly with the latest GNN architectures. We deliberately chose to compare KA-GAT with classical GNN models (e.g., GCN, GAT) and existing KAN-GNN hybrids to isolate and evaluate the contributions of the KA layer in graph learning tasks.
> > >
> > > That said, we agree that comparisons with recent GNNs, such as GATv2 and APPNP, would provide valuable insights and strengthen the evaluation. These models are excellent candidates for future experiments, particularly as we explore how the KA layer can enhance or complement their mechanisms.
> > >
> > > **3.Lack of Solid Theoretical Analysis**
> > >
> > > We appreciate the importance of rigorous theoretical analysis and acknowledge that this aspect of the current manuscript could be further developed. Our initial analysis focused on demonstrating how the Kolmogorov-Arnold decomposition enhances feature representation by decomposing nonlinear relationships into simpler components. While this provides a conceptual foundation, we recognize that a more formal mathematical treatment is needed to fully articulate the expressive power and generalization capabilities of KA-GAT.
> > >
> > > Moving forward, we aim to build on methodologies from recent works in ICLR, NeurIPS, and ICML to establish a robust theoretical framework for KA-GAT. This will include analyzing its capacity to generalize across diverse graph structures and tasks.
> > >
> > > **Positioning of the Work and Future Plans**
> > >
> > > This work represents a first step in exploring the integration of Kolmogorov-Arnold layers with GNNs. We believe the proposed KA-GAT model demonstrates clear advantages in terms of feature decomposition and interpretability, as supported by both experimental results and visualizations in the revised manuscript. While we acknowledge that broader experiments and deeper theoretical analysis are essential for future extensions, the current study successfully establishes a foundation for further research in this direction.
> > >
> > > Your thoughtful feedback has been instrumental in identifying areas for improvement, and we sincerely thank you for your insights. We are committed to building on this work and addressing these open questions in future studies.
> > >
> > > Best regards

---

### Note · Authors · 2025-01-02

I have read and agree with the venue's withdrawal policy on behalf of myself and my co-authors.